# Cost-Effectiveness Analysis of Dabrafenib Plus Trametinib and Vemurafenib as First-Line Treatment in Patients with BRAF V600 Mutation-Positive Unresectable or Metastatic Melanoma in China

**DOI:** 10.3390/ijerph18126194

**Published:** 2021-06-08

**Authors:** Tianfu Gao, Jia Liu, Jing Wu

**Affiliations:** 1School of Pharmaceutical Science and Technology, Tianjin University, Tianjin 300072, China; tianfugao@tju.edu.cn (T.G.); jial@tju.edu.cn (J.L.); 2Center for Social Science Survey and Data, Tianjin University, Tianjin 300072, China

**Keywords:** dabrafenib plus trametinib, vemurafenib, cost-effectiveness, melanoma, China

## Abstract

*Objective:* To evaluate the cost-effectiveness of dabrafenib plus trametinib combination therapy versus vemurafenib as first-line treatment in patients with BRAF V600 mutation-positive unresectable or metastatic melanoma from a healthcare system perspective in China. *Methods:* This study employed a partitioned survival model with three health states (progression-free survival, post-progression survival and dead) to parameterize the data derived from Combi-v trial and extrapolated to 30 years. Health states’ utilities were measured by EQ-5D-3L, also sourced from the Combi-v trial. Costs including drug acquisition costs, disease management costs and adverse event costs were based on the Chinese Drug Bidding Database and physician survey in China. The primary outcomes of the model were lifetime costs, life-years (LYs), quality-adjusted life-years (QALYs) and incremental cost-effectiveness ratio (ICER). Deterministic and probabilistic sensitivity analyses were conducted, respectively. *Result:* Dabrafenib plus trametinib is projected to increase a patient’s life expectancy by 0.95 life-years over vemurafenib (3.03 vs. 2.08) and 1.09 QALY gains (2.48 vs. 1.39) with an incremental cost of $3833. The incremental cost-effectiveness ratio (ICER) was $3511 per QALY. In the probabilistic sensitivity analyses, at a threshold of $33,357 per QALY (three times the gross domestic product (GDP) per capita in China in 2020), the probability of dabrafenib plus trametinib being cost-effective was 90%. In the deterministic sensitivity analyses, the results were most sensitive to the dabrafenib plus trametinib drug costs, vemurafenib drug costs and discount rate of cost. *Conclusion:* Dabrafenib plus trametinib therapy yields more clinical benefits than vemurafenib. Using a threshold of $33,357 per QALY, dabrafenib plus trametinib is very cost-effective as compared with vemurafenib in China.

## 1. Introduction

Melanoma is an aggressive type of malignant tumor arising from melanocytes, typically in the skin, and presents severe clinical, economic and societal burden [1]. Melanoma is a relatively common malignancy in the West, but has a lower incidence in Asians [2,3]. In 2018, global new cases of melanoma of skin were estimated to be 287,723, accounting for 1.6% of all sites of cancer, and 60,712 cases died, accounting for 0.6% of all cancers [4]. Melanoma is relatively rare in China, which had an estimated 8000 new cases and 3200 deaths in 2015 [5]. However, over 40% of melanoma patients in China are diagnosed with locally advanced disease (stage III and IV) and prognosis is poor, with the 5-year survival rate for stage IV patients being only 4.6% [6].

For many years, Dacarbazine (DTIC) was the only first-line treatment approved for advanced or metastatic melanoma in China, despite the limited clinical benefits [7,8]. Current systemic treatments for patients with previously untreated advanced or metastatic melanoma in China include immunotherapy, targeted therapy and chemotherapy. Targeted therapy can be used in patients with advanced melanoma with activated mutations in BRAF, which is a constituent of the mitogen-activated protein kinase (MAPK) pathway. Approximately 25.5% of melanomas in Chinese patients harbor BRAF V600 mutations [9]. Vemurafenib and dabrafenib plus trametinib are highly selective small-molecule inhibitors that have antitumor effects against melanoma with mutant BRAF kinase in the MAPK pathway [10,11].

Combi-v (ClinicalTrials.gov, accessed on 28 April 2021, identifier: NCT01597908) is a phase 3, open label, double-arm, randomized-controlled clinical trial evaluating the effect of first-line dabrafenib (150 mg orally twice daily) and trametinib (2 mg orally once daily) combination therapy with vemurafenib (960 mg orally twice daily) monotherapy on overall survival (OS) and progression-free survival (PFS) in previously untreated patients with unresectable stage IIIC or IV melanoma with BRAF V600E or V600K mutations (V600E/K accounts for 90% in V600) started from June 2012. A total of 704 patients were well balanced in the two groups, with a baseline median age of 55 and a male sex percentage of 55% [12,13]. At the updated data cut-off (13 March 2015), dabrafenib and trametinib significantly improved OS over vemurafenib (hazard ratio, 0.66; 95% confidence interval [CI], 0.53–0.81; *p* < 0.001) and median OS of 25.6 vs. 18 months. A significant improvement also observed in PFS for dabrafenib plus trametinib compared with vemurafenib (hazard ratio, 0.61, 95% CI: 0.51–0.73, *p* < 0.001) and median PFS of 12.6 vs. 7.3 months, respectively [14].

In March 2017, single-agent BRAF inhibitor vemurafenib (Roche, Basel, Switzerland) was the first targeted therapy approved for treating advanced melanoma patients with BRAF V600 mutation by the National Medical Products Administration (NMPA) in China and was included into the National Reimbursement Drug List (NRDL) after the price negotiation by the National Healthcare Security Administration (NHSA) in 2017. Until recently, BRAF and MEK inhibitors dabrafenib plus trametinib (Novartis, Basel, Switzerland) combination targeted therapy was approved for the same indication at the end of 2019 and was successfully included in the NRDL after the price negotiation by the NHSA in 2020.

To our knowledge, since dabrafenib and trametinib were just added into the NRDL, cost-effectiveness evidence is needed using the updated NRDL-negotiated price to provide a reference for clinical decision-making. To address this need, we compared the cost-effectiveness of dabrafenib plus trametinib with vemurafenib as the first-line treatments for previously untreated patients with unresectable stage IIIC or IV melanoma with BRAF V600 mutations from a healthcare system perspective in China.

## 2. Methods

### 2.1. Overview

A partitioned survival model was developed to assess the expected progression-free survival, overall survival, quality-adjusted life-years (QALYs) and lifetime costs between dabrafenib plus trametinib and vemurafenib. The target patient population consisted of patients with BRAF V600 mutation positive unresectable or metastatic melanoma, who had no prior systemic anti-cancer therapy, corresponding to the patients in the Combi-v trial [12]. The time horizon was 30 years, starting from patients entering the model. This analysis was conducted from a Chinese healthcare system perspective; thus, only direct healthcare costs related to treatment of metastatic melanoma were considered. The effectiveness, i.e., the health outcomes, was assessed in terms of life-years (LYs) and QALYs. The primary economic outcome is the incremental cost-effectiveness ratio (ICER). In the base case analysis, both cost and effectiveness were discounted at a rate of 5% per year, as recommended by the China guidelines for pharmacoeconomic evaluations [15]. Cost parameters were converted to 2020 United States dollars (1 dollar = 6.52 Chinese yuan). The model was developed and run in Microsoft^®^ Excel 2016.

This study used the clinical data obtained in the Combi-v to perform a cost-effectiveness study and does not contain any studies with human participants or animals performed by any of the authors. The Combi-v randomized controlled trial (RCT) aims to compare the efficacy of dabrafenib plus trametinib and vemurafenib at 193 centers world-wide from 2012 to 2019. This RCT was conducted in accordance with the provisions of the Declaration of Helsinki and Good Clinical Practice guidelines. The protocol was approved by the institutional review board or human research ethics committee at each study center and registration at Clinicaltrials.com, accessed on 28 April 2021, number NCT01597908, available at https://www.nejm.org/doi/full/10.1056/NEJMoa1412690 (accessed on 27 April 2021).

### 2.2. Model Structure

The partitioned survival model is similar to that used in prior economic evaluations of treatments for advanced or metastatic cancers with three mutually exclusive health states: progression-free survival (PFS), post-progression survival (PPS) and dead (Figure 1) [16,17]. Patients were set into a progression-free state when entering the model and received targeted therapy until detection of progression. The proportion of patients in each health state over the course of time was estimated based on the survival distribution for PFS and OS survival. Considering the cycle duration of the two therapies, the cycle length of the model was set at 1 week, which could also eliminate the need for a half-cycle correction. Costs and QALYs were dependent on the treatments and the expected time in each health state. Expected PFS and OS time were estimated as the area under the curve (AUC) for the PFS and OS curves, respectively. Expected PPS was calculated as the area between the expected PFS and OS curves. This model allowed for the consideration of one-off costs associated with treatment initiation, adverse events (AEs), post treatment anticancer therapy and death.

### 2.3. Estimation of Progression-Free Survival and Overall Survival

The modeling of PFS and OS were combined by two main segments: the trial period and projection period. The trial period of PFS and OS for dabrafenib plus trametinib and vemurafenib were estimated based on the survival data from the 13 March 2015 cut-off date of Combi-v [14]. PFS and OS parametric distributions in each arm were estimated by fitting restructured individual patient-level data from Combi-v. Six parametric distributions were considered: exponential, Weibull, log-logistic, log-normal, Gamma and Gompertz. The goodness of fit of the distributions was evaluated based on visual inspection, Akaike information criterion (AIC) and Bayesian information criterion (BIC). In the PFS estimation, although the Gamma distribution generally provided the best AIC fit, it tended to generate a long-tailed curve that might overestimate the PFS time in both treatment groups. Log-normal distribution was selected based on the best visual inspection and the best BIC statistic fit for both arms (Figure 2) (Table A1, Figure A1 and Figure A2 in Appendix A). OS for dabrafenib plus trametinib and vemurafenib during the trial period were available at 30.5 months and 29.5 months. Log-normal and log-logistic were selected for the OS trial period with the best fit for dabrafenib plus trametinib and vemurafenib, respectively. A 30-year projection of OS based on parametric curves fit to trial period OS data is shown in Figure 3 (Table A2, Figure A3 and Figure A4 in the Appendix A).

### 2.4. Utility Estimates

Health state utility values were estimated based on the EuroQol-5D-3L (EQ-5D-3L) data reported from the two treatment arms in the Combi-v trial [18]. In the trial, the mean baseline EQ-5D utility scores were similar between treatment groups (0.751 for combination therapy vs. 0.715 for vemurafenib). Changes from baseline utility values were measured and reported for the follow-up assessment weeks. The mean changes from baseline utility values for the PFS state was 0.072 for patients receiving dabrafenib plus trametinib in combination and −0.027 for the vemurafenib group; the mean changes from baseline utility values for the PPS state was 0.06 and −0.05 for each group, respectively [18]. Therefore, PFS utilities of 0.823 and 0.688 were assigned to patients receiving combination therapy and vemurafenib, respectively; PPS utilities of 0.811 and 0.665 for combination therapy and vemurafenib, respectively.

### 2.5. Costs Estimates

Costs were estimated from healthcare system perspective; thus, only direct medical costs were included. The medical costs considered in this model included drug acquisition costs, treatment initiation costs, disease management costs, adverse events (AEs) costs, post treatment anti-cancer therapy (PTACT) costs and terminal care costs. Drug acquisition costs were estimated from the Chinese Drug Bidding Database by the MENET (www.menet.com.cn, accessed on 20 April 2020) and price assumption, while other costs of monitoring and follow-up were obtained from physician surveys in five tertiary hospitals located in Beijing, Tianjin, Chengdu, Shanghai and Guangzhou of China.

Drug acquisition costs were estimated based on dosing schedule reported in Combi-v [12]. Dabrafenib and trametinib were newly added into the NRDL in 2020. In this study, dabrafenib plus trametinib drug acquisition costs were based on the latest reimbursement price set by NHSA in December 2020. The daily drug costs of dabrafenib plus trametinib is $113.4. Vemurafenib was added into the NRDL in 2017. The latest drug cost of vemurafenib ($110 per day) was assumed to be 20% cut of its original list price ($137.5 per day) after the 2020 NRDL negotiation in the base case.

Treatment initiation costs were estimated as a one-time cost associated with all examination services needed to diagnose melanoma before initiating the targeted therapy. Monthly disease management costs incurred in the progression-free and post-progression states included outpatient physician visit costs, hospitalization costs and laboratory tests costs. The frequencies of resource use were assumed to be the same for both arms by health states. The utilization frequency and unit price were obtained from physician surveys.

The model included the cost of Grade 3–5 AEs with an incidence of 5% or more reported in the most recently published report of five-year outcomes of Combi-v [13], which included hypertension, pyrexia, rash, neutropenia, y-glutamyltransferase (GGT) and squamous cell carcinoma. Per-event costs were obtained from the physician survey.

Post treatment anti-cancer therapy (PTACT) costs were applied to both treatment arms who entered the progressed disease state. Costs included the acquisition costs associated with targeted therapy, immune checkpoint inhibitors and chemotherapy. Mean PSACT costs were used in the model and estimated based on physician surveys. Terminal care costs were also included as a one-time cost in the model. Utilities and costs inputs in the model are listed in Table 1.

### 2.6. Sensitivity Analyses

Sensitivity analyses were conducted to evaluate the uncertainty and robustness of the model. Deterministic sensitivity analyses (DSA) were estimated to assess the impact of key parameters on model outputs. Probabilistic sensitivity analyses (PSA) were performed with 1000 iterations of Monte Carlo simulations. All parameters were varied with a defined statistical distribution simultaneously. Gamma distribution was used for all input costs, and a beta distribution was adopted for utility and discount rate. The results of DSA and PSA were represented as a tornado diagram and cost-effectiveness acceptability curves (CEAC), respectively.

## 3. Results

### 3.1. Base Case Results

In the base case, dabrafenib plus trametinib (D + T) yields 3.03 LYs compared with 2.08 LYs in the vemurafenib group, providing 0.95 more LYs survival benefit (0.78 more progression—free life years and 0.17 more post-progression life years). The combination group obtained 1.09 more QALYs than vemurafenib (2.48 QALYs vs. 1.39 QALYs). Total life-time costs were $119,766 and $115,933 in the D + T group and vemurafenib group, respectively. Medication costs were $74,145 in the D + T group and $40,571 in the vemurafenib group, yielding a cost difference of $33,574. For the disease management costs, in the combination group this was $2662 higher in the PFS state and $1066 higher in the PPS state. PSACT costs for the D + T group were $33,114 less than vemurafenib. The ICER for D + T versus vemurafenib was $3511 per QALY gained (Table 2).

### 3.2. Sensitivity Analyses

The results of DSA are presented in a tornado diagram (Figure 4). The drug costs of dabrafenib plus trametinib had the greatest influence on the ICER, followed by the vemurafenib drug cost and discount rate of the costs. The PSA results showed that the D + T group yields more QALYs in 99% of simulations. The cost-effectiveness acceptability curve for D + T versus vemurafenib shows that when the willingness-to-pay is $33,357 per QALY (three times the GDP per capita in China in 2020), the probability of D + T being cost-effective is 90% [19]. At a threshold of $22,238 per QALY (two times GDP per capita in China in 2020), the probability of D + T being cost-effective is 82%. At a threshold of $11,119 per QALY (one times the GDP per capita in China in 2020), the probability of D + T being cost-effective is 66% (Figure 5 and Figure 6).

## 4. Discussion

On 28th December 2020, the NHSA released the official results of the latest NRDL update. There were 162 drugs on the shortlist, and 119 of them eventually made the listing with a success rate of 73%; price cuts averaged at 51% for the drugs made through negotiations. Dabrafenib and trametinib were recently approved at the end of 2019 in China and were successfully included in the NRDL after the price negotiation by the NHSA in 2020. To our knowledge, this study is the first analysis after the 2020 NRDL updated to evaluate the health and economic outcomes of dabrafenib plus trametinib treatment as a first-line treatment for patients with BRAF V600 mutation-positive unresectable or metastatic melanoma in China.

A partitioned survival model was developed to evaluate the cost-effectiveness of dabrafenib plus trametinib versus vemurafenib as a first-line treatment for patients with BRAF V600 mutation-positive unresectable or metastatic melanoma from a Chinese healthcare system perspective. The results showed that dabrafenib plus trametinib combination treatment group obtained a survival benefit of 0.95 LYs and 1.09 QALYs over the vemurafenib treatment group with an additional cost of $3833. With a willingness-to-pay threshold of $33,357 per QALY (three times the GDP per capita in China in 2020), the ICER is less than 1 GDP per capita, and hence, dabrafenib plus trametinib therapy is very cost-effective compared to vemurafenib after the NRDL price negotiation in 2020. Deterministic sensitivity analyses revealed that the ICER was most sensitive to the drug cost. The probabilistic sensitivity analyses showed that at the $33,357 per QALY threshold, the probability of dabrafenib plus trametinib being more cost-effective than vemurafenib is 90%.

There was only one published cost-effectiveness comparison between dabrafenib plus trametinib and vemurafenib as a first-line treatment in patients with BRAF V600 mutation-positive unresectable or metastatic melanoma, which was in a Swiss setting [20]. The Swiss-based analysis showed that dabrafenib plus trametinib conferred greater health benefits with a 0.52 QALY gained at an additional cost of Swiss Francs CHF199,647 compared with vemurafenib. Three main reasons could explain the difference in QALYs between the Swiss-based study and our analysis. Firstly, the clinical data of the Swiss-based study was based on the Combi-v trial, which had a the cut-off date of 17 April 2014. Our analysis adopted a longer follow-up period of 30.5 months for overall survival from the March 2015 data cut-off date of Combi-v. Secondly, the utility value of D + T in the Swiss setting was assumed to be the same as that for trametinib monotherapy with the reason of unclear if D+T could result in other utilities. This utility assumption will underestimate the clinical benefits of D + T therapy, whereas for our analysis, the treatment-specific utility values were based on EQ-5D-3L data reported in the head-to-head Combi-v. Lastly, the Swiss-based study employed a Markov model structure, assuming constant transition probabilities converted from hazard rates. Our analysis adopted a partitioned survival model using a parametric distribution to fit the PFS and OS clinical data. Therefore, the different length of the follow-up period, utility values and model settings could be the reasons of the difference in QALY gains in the two analyses.

This study is subject to a number of limitations. Firstly, the clinical trial data and utility values were based on Combi-v because Chinese population evidence was lacking; these data may not be generalizable for Chinese patients. Secondly, currently, there is a lack of data on the economic burden for patients with BRAF V600 mutation-positive metastatic melanoma in China. The costs of monitoring and follow-up were obtained from a physician survey in five tertiary hospitals located in five large cities in China; thus, the limited number of survey hospitals may result in a bias in the costs estimates. Thirdly, utility values in this study were based on EQ-5D-3L data from Combi-v. EQ-5D assessments were not conducted for a long follow-up period beyond progression; thus, utility values of the disease progression may not reflect HRQoL for the entire post-progression period. Given these uncertainties, a number of sensitivity analyses were conducted and it has shown that the model results are robust. Lastly, because maximum follow-up data was only 30.5 months in the Combi-v, it was necessary to project PFS and OS beyond the end of the follow-up using parametric survival distributions fit to the data from Combi-v. These projections are associated with substantial uncertainty, which may impact the cost-effectiveness estimates. Given this unpredictability around the extrapolation of patient survival, the model’s results should be validated against the long-term efficacy data or against a larger sample size with real-world evidence as the data be available.

PFS and PPS utility values were estimated from the mean changes from baseline utility values reported in the Combi-v. The D + T baseline utility value is 0.751, while that of vemurafenib is 0.715; thus, there is a 0.036 baseline utility benefit in the D + T group. Among patients receiving D + T therapy, health utility mean scores increased relative to baseline scores for all follow-up assessment weeks, including at disease progression, 0.072 mean changes for PFS and 0.06 changes for PPS. For patients receiving vemurafenib, the health utility was unchanged or decreased for all assessment weeks with a −0.027 mean change for PFS and −0.05 for PPS. There are two reasons that could be considered when interpreting these results. First, patients and investigators were not masked to the treatment group, and patients’ expectations about treatment efficacy might affect their responses. Second, patients participating in Combi-v might be more motivated and optimistic in the D + T group, and therefore, more likely to report benefit and endure more treatment-related toxicity than the vemurafenib group. However, it is worth noting that besides the EQ-5D assessment, the European Organization for Research and Treatment of Cancer quality of life (EORTC QLQ-C30) and the Melanoma Subscale and the Functional Assessment of Cancer Therapy-Melanoma (FACT-M) were also employed in Combi-v, which have consistently shown that dabrafenib plus trametinib adds a clear benefit over vemurafenib for HRQoL. The consistency of these three HRQoL assessments supports the robustness of the results.

## 5. Conclusions

From a Chinese healthcare system perspective, dabrafenib plus trametinib therapy yields more clinical benefits with lower medical costs over vemurafenib. Using a threshold of $33,357 per QALY, dabrafenib in combination with trametinib is a very cost-effective treatment option compared with vemurafenib for previously untreated patients with BRAF V600 mutation-positive unresectable or metastatic melanoma in China. The results of this study may serve as an important reference for dabrafenib plus trametinib treatment decision-making in China and offer scientific guidance and reference for other countries to conduct their own melanoma cost-effectiveness analysis.

## Figures and Tables

**Figure 1 ijerph-18-06194-f001:**
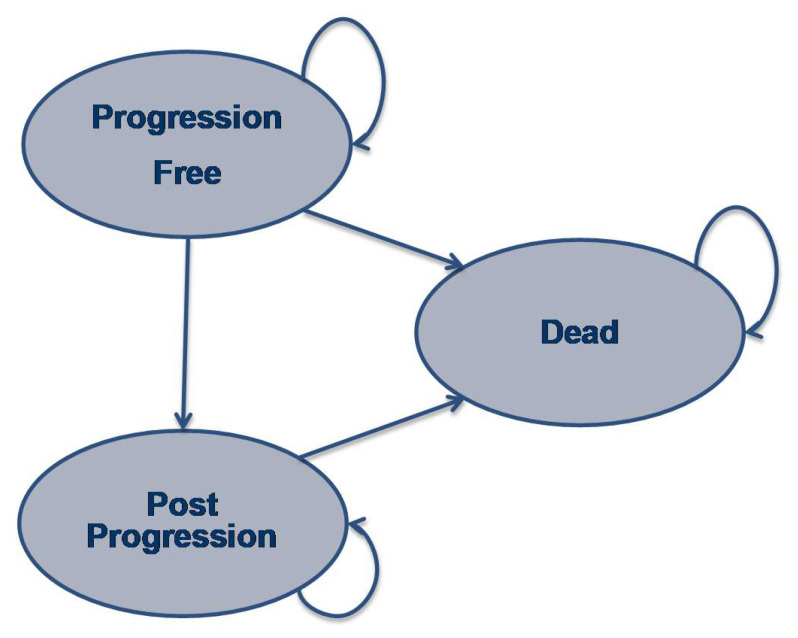
Partitioned survival model.

**Figure 2 ijerph-18-06194-f002:**
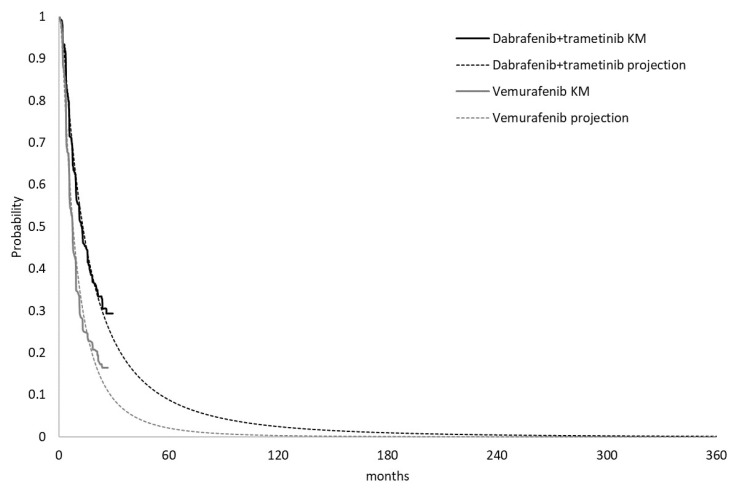
Kaplan-Meier (KM) and projection curve of progression free survival.

**Figure 3 ijerph-18-06194-f003:**
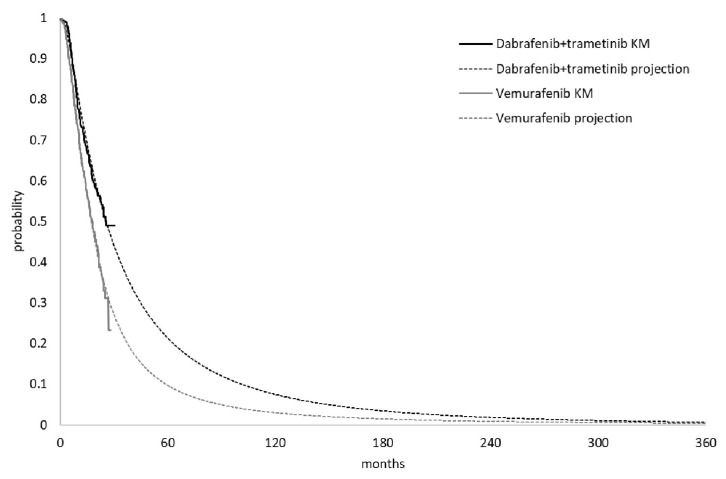
Kaplan-Meier (KM) and projection curve of overall survival.

**Figure 4 ijerph-18-06194-f004:**
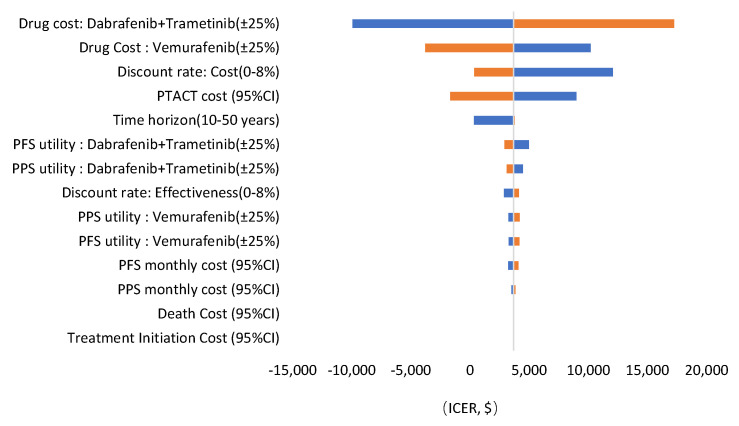
Tornado diagram.

**Figure 5 ijerph-18-06194-f005:**
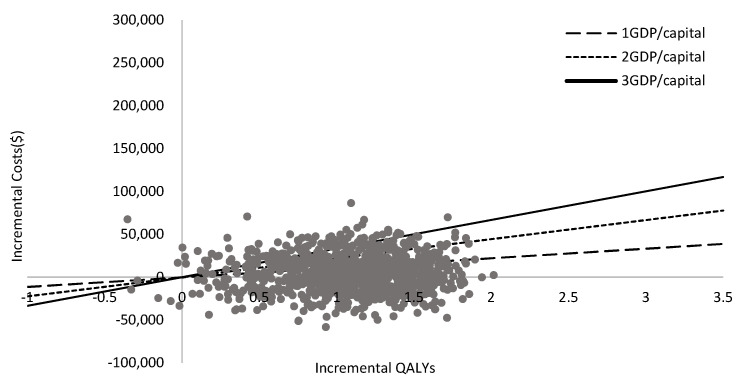
Probabilistic sensitivity analyses.

**Figure 6 ijerph-18-06194-f006:**
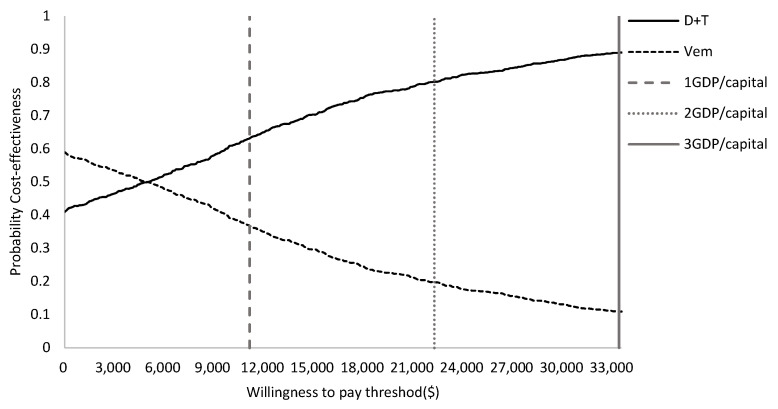
Cost-effectiveness acceptability curve.

**Table 1 ijerph-18-06194-t001:** Utilities and costs inputs.

Model Parameter	D + T	Vemurafenib	Source/Reference
Utility values			Grob et al. [18]
Progression-free	0.823	0.688	
Post-progression	0.811	0.665	
Daily cost of drug ($)	113	110	MENET/Price assumption
Monthly costs of disease management ($)			Physician survey
Progression-free	284	284	
Post-progression	528	528	
One-time costs ($)			Physician survey
Treatment initiation	2685	2685	
PSACT	25,961	59,710	
Terminal care	6139	6139	
Adverse events costs per treatment ($)			Physician survey
Hypertension	105		
Pyrexia	123		
Rash	138		
Neutropenia	87		
GGT	167		
SCC	307		

D + T, Dabrafenib plus Trametinib; PTACT, Post-treatment Anti-Cancer Therapy; GGT, y-glutamyltransferase; SCC, squamous cell carcinoma.

**Table 2 ijerph-18-06194-t002:** Base case results.

	D + T	Vemurafenib	Incremental
Costs, $, Discounted			
Medications	74,145	40,571	33,574
Adverse events	47	114	−66
Treatment initiation	2685	2685	0
PFS disease management costs	6107	3445	2662
PPS disease management costs	7849	6783	1066
Costs of Post-treatment Anti-Cancer Therapy	23,705	56,820	−33,114
Terminal care	5227	5515	−288
Total	119,766	115,933	3833

LYs, Discounted			
PFLYs	1.79	1.01	0.78
PPLYs	1.24	1.07	0.17
LYs	3.03	2.08	0.95

QALYs	2.48	1.39	1.09

ICER($/QALY)			3511

PTACT, Post-treatment Anti-Cancer Therapy; PFLYs, Progression-free life years; PPLYs, Post-progression life years; LYs, Life years; QALYs, Quality-adjusted life-years; ICER, Incremental cost-effectiveness ratio.

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
