# Peer review of "Cost-Effectiveness Analysis of Dabrafenib Plus Trametinib and Vemurafenib as First-Line Treatment in Patients with BRAF V600 Mutation-Positive Unresectable or Metastatic Melanoma in China"

_ijerph, 2021, doi:10.3390/ijerph18126194_

Round 1

Reviewer 1 Report

Dear authors,

I appreciate the opportunity to read your article on a very current and interesting topic.
However, I have some concerns about ethical issues and conflict of interest.
I would ask you to include in the methods of the article the approval of the study by the ethics committee, as well as the justification why despite Novartis being the sponsor of the study, the company logo does not appear on the study materials, such as the questionnaire.
Regarding conflict of interest, I think it needs to be clarified.

Author Response

Thanks for your valuable comments. Please see the attachment.

Reviewer 2 Report

The manuscript by Gao et al. entitled “Cost-effectiveness analysis of Dabrafenib plus Trametinib and Vemurafenib as first line treatment in patients with BRAF V600 mutation-positive unresectable or metastatic melanoma in China” concerns analysis of the costs of the treatment of melanoma with two small molecule kinases inhibitors. Authors analysed and calculated data for China. Authors compared the dabrafenib plus trametinib regimen with vemurafenib treatment. Several limitations of the study were listed (e.g. clinical trial data and utility values were based on Combi-v study, limited number of survey hospitals evaluated for the costs of monitoring and follow-up), but at the same time Authors conducted a number of sensitivity analyses. Authors demonstrated advantages of dabrafenib plus trametinib treatment when compared to vemurafenib in term of patient’s life expectancy and quality-adjusted life-years. The results of the study might be useful while making the decision on the melanoma therapy.

In my opinion, the manuscript could be published in IJERPH in current form.    

Author Response

(The authors gave the same response as above.)

Reviewer 3 Report

This article entitled “Cost-effectiveness analysis of Dabrafenib plus Trametinib and Vemurafenib as first line treatment in patients with BRAF V600 mutation-positive unresectable or metastatic melanoma in China” by Tianfu Gao, et al. is a cost-effectiveness analysis using estimated cost in China. The subject of this study will be of interest to readers of International Journal of Environmental Research and Public Health, however, I have the following concerns on the current form.

Comments

1. Why is there a big difference in costs of post-treatment anti-cancer therapy between dabrafenib plus trametinib and vemurafenib? Possible causes should be discussed in Discussion.

2. Are the results of this study possibly applied to any other countries? Are the results true only in China? The authors are encouraged to refer to the general applicability of their results.

Author Response

(The authors gave the same response as above.)
